# Have Sustainable Development Goal Depictions Functioned as a Nudge for the Younger Generation before and during the COVID-19 Outbreak?

**Takuro Uehara** *  and **Ryo Sakurai** 

College of Policy Science, Ritsumeikan University, Osaka 567-8570, Japan; ryo223sak@gmail.com
* Correspondence: takuro@fc.ritsumei.ac.jp; Tel.: +81-72-665-2080

**Abstract:** Sustainable Development Goals (SDGs) and their corresponding logos have become ubiquitous in Japan. While not legally binding, they allow us to choose how to contribute or not to the SDGs. Considering that SDGs share characteristics with nudges, we investigated whether SDGs, with their term and logos, have functioned as a nudge before and during the COVID-19 outbreak. Using Japan as a case study, we analyzed newspaper articles to explore how the term SDGs has spread before and during the outbreak. We also conducted a questionnaire among college students (n = 421) to explore how exposure to the term or its logos has steered the behavior of the younger generation toward SDGs. Our analysis revealed that the use of the term in newspaper articles has rapidly increased and spread across newspaper sections, whereas the COVID-19 outbreak has slowed its spread. The results showed that 68.9% of the respondents were familiar with the term or logos. Of these, 25.4% had changed their behavior toward SDGs. Surprisingly, COVID-19 has had a rather positive influence as a catalyst in that more respondents have overall become more proactive or maintained previous behaviors (28.3%). This indicates that COVID-19 may be an opportunity to make a shift toward a more sustainable society.

**Keywords:** SDGs; nudge; content analysis; COVID-19; sustainable development goals; college students

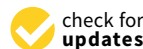



## 1. Introduction

The 17 Sustainable Development Goals (SDGs) adopted by the United Nations (UN) General Assembly in September 2015 are an integral part of the 2030 Agenda for Sustainable Development [1]. The abbreviation "SDGs" and their logos are ubiquitous in Japan. SDGs are promoted by government entities at the local and national levels, as well as by other entities. Citizens place SDGs logo badges on business suits. Trains wrapped with adverts promoting SDGs are in operation. The mass media covers stories on how we can change our behavior to be consistent with the SDGs. Companies promote their support for SDGs on their websites and with their products. Therefore, we assume that college students in Japan are well-exposed to SDGs. Whether they notice them and change their behavior accordingly must be investigated. For these reasons, we aimed to discover if the SDG term and logos can function as a nudge for steering people's behavior toward SDGs.

A number of studies have focused on impact assessments of SDGs toward sustainable development. As SDGs comprise goals, targets, and indicators, major studies have assessed the progress toward sustainable development by employing a variety of indicators relevant to the SDGs [2–4]. Schmidt-Traub et al. [5] revealed that their SDG index was correlated with other key indicators for sustainable development, including per capita gross domestic product, the Human Development Index, and subjective well-being. As SDGs essentially embrace trade-offs [6], some studies combine SDG indicators with other methods to capture the interdependencies [7]. Other studies link SDGs to sustainable development by employing other methods. Costanza et al. [8] proposed an approach that combines

SDGs with a Sustainable Wellbeing Index (SWI) using a system dynamics model. This approach reflects and synthesizes the non-linear dynamic interdependencies of SDGs and assesses the overall progress of sustainable development using a system dynamics model [8]. However, to our knowledge, this is the first study to investigate the impact of the SDG term and their logos as a nudge.

A nudge is a term used in behavioral economics to describe a public or private intervention that alters human behavior in particular directions, but allows them to act without forbidding options or significantly changing material incentives [9–11]. While nudges are hardly new and are built into the fabric of human society, they are a recent trend in evidence-based policymaking, drawing insights from behavioral economics as a substitute or addition to more coercive approaches, such as command and control regulations [10,12–14]. Nudges play a large role in the United States in multiple areas, including environmental protection [14]. The Ministry of the Environment of Japan launched the Behavioral Sciences Team to promote nudges [15]. Nudges have caught the attention of policymakers and researchers for practical and philosophical reasons, rendering them a novel approach. Their benefits are that they often have large effects with low costs and they preserve both human agency and freedom of choice [9]. In other words, to qualify as a nudge, it must be cost effective and easy to avoid, but not mandatory [10]. For example, simply adding information to a consumer's monthly bills contributes to energy savings ranging from 2% to 6% [16]. In addition, a wide range of actors can nudge from governments to businesses to individuals, in contrast to command and control regulations [10].

Nudges can take various forms. They can be one of two types according to their purpose: assisting individuals to make choices that are in their own best interest (e.g., saving more for retirement), referred to as "type 1 nudges," or steering an individual's behavior to achieve a desired collective end (e.g., encouraging environmentally friendly practices), referred to as "type 2 nudges" [10]. According to their level of intrusion, nudges can be divided into five types: information dissemination (e.g., government campaigns), governmentally mandated information dissemination (e.g., calorie labels), setting default rules (e.g., defaulting air travelers into the payment of carbon offsets), manipulation (e.g., subliminal advertising), and other mandates [17]. Nudges may change behavior by helping people counteract a variety of biases and heuristics (rules of thumb) or inform them in the absence of bias (i.e., they may lack information in the first place) [9,18]. For example, there may be attempts to counteract a present bias by encouraging people to pay attention to long-term consequences [9]. Nudges are considered appropriate or needed when decisions "are difficult and rare, for which they do not get prompt feedback, and when they have trouble translating aspects of the situation into terms that they can easily understand" [11].

SDGs share their characteristics with nudges in four aspects. First, while SDGs are grounded in international law, they are neither legally binding nor intended to grant immediate legal force to the goals [19,20]. Therefore, people and organizations can decide how much they contribute toward the goals. In other words, while they are indeed encouraged to contribute to SDGs, people are not directed as to what to do. While our society has not remained on track for all goals, likely due to this non-binding structure [2,21], they provide an opportunity to function as a nudge. Second, SDGs involve issues that are highly complex; moreover, it may be difficult to receive prompt feedback for actions. With 17 goals, 169 targets, and 231 indicators proposed, SDGs involve complex interconnections [8]. Furthermore, we cannot often assume that individual actions aggregate up to system-level sustainability [22]. In other words, what individuals contribute toward sustainability does not necessarily lead to a sustainable society. As sustainability involves long-term and wide-ranging system change, it takes time to witness improvements caused by our behavioral changes. Third, SDGs can be a type 2 nudge with the level of intrusion of information campaigns steering an individual's behavior to achieve a desired collective end [10,17]. Fourth, contrary to command and control regulations, various actors (government, businesses, and individuals) have promoted SDGs in various forms. The UN Secretary-General called on all sectors of society to take part in this challenge [1]. For example, a survey

revealed that 24.4% of businesses in Japan are implementing or willing to contribute to SDGs [23]. Some not only implement, but also enlighten their customers. For example, a train company decorated its trains with SDGs as part of an information campaign [24]. There are media feature articles that propose how the Japanese can contribute to SDGs, along with a reflection on indigenous culture [25].

The recent outbreak of COVID-19 has led us to pose an additional question: How does the outbreak impact behaviors toward SDGs? Can this be an opportunity to change or is it a risk to achieving SDGs? This may be an opportunity, as claimed by the United Nations: "Leveraging this moment of crisis, when usual policies and social norms have been disrupted, bold steps can steer the world back on track toward the Sustainable Development Goals. This is the time for change, for a profound systemic shift to a more sustainable economy that works for both people and the planet" [26]. During periods of turmoil, SDGs can be at risk because policies deemed non-essential may be postponed [27]. Businesses and individuals may also be busy adapting to the current situation, leaving little room to pay attention to SDGs.

The purpose of this study was to assess if SDGs, as a term or logo, have functioned as a nudge or a choice architecture before and during the COVID-19 crisis. While there have been studies examining the use of nudges to achieve SDGs [28,29], to our knowledge, this is the first study on how the SDG term or logos nudge people to change their behaviors to contribute to achieving SDGs. As SDGs are a proper noun, i.e., not a common noun, such as "sustainability," they may be more noticeable to the public. Logos related to SDGs are also often used to promote SDGs. Therefore, they can nudge people, such as the function of eco-labels, although they have not always been successful [30,31]. Given its potential impacts on sustainability goals, investigating the impact of COVID-19 is an urgent need; however, few studies have reported on this.

We pose three research questions to assess if the SDG term and its logos function as nudges.

Research question 1: How much have SDGs permeated Japanese society?
Research question 2: By how much do SDGs reach and influence younger generations in Japan via its term and logos?
Research question 3: How has COVID-19 impacted the spread of SDGs and the attitudes of younger generations toward SDGs in Japan?

To answer these questions, we reviewed the news coverage of SDGs in a major newspaper and assessed the attitudes of college students toward SDGs in Japan. Newspapers are a type of major mass media that can reach a broad range of people and reflect on and shape public discourse [32,33]. The media stores memories, maps where we are and who we are, and provides materials to guide us into the future [33]. There are various theories and models for mass media effects, such as the classic stimulus–response model [33]. While they have traditionally been published in print, most newspapers are now also published on their official websites or provided via portal websites. The younger generation, including college students, are critical in establishing a shift toward a sustainable society. They are mature enough to learn and ready to steer, if they wish, their behavior or career path to be a leader, policy maker, scientist, consumer, researcher, or an entrepreneur who contributes to the attainment of SDGs [34].

The remainder of this paper is structured as follows. Section 2 addresses the materials and methods used for the study. Section 3 reports and discusses the results. Section 4 concludes this paper.

## 2. Materials and Methods

### 2.1. Content Analysis

We used the Yomiuri Database Service (https://database.yomiuri.co.jp/rekishikan/ (accessed on 3 February 2021)) to search for articles that included the abbreviated term "SDGs" in the main text or title from January 2015 through June 2020. The database covers news articles that appeared in the Yomiuri Shimbun, one of the most subscribed

to newspapers in Japan. Eight million papers are sold every day, covering 13.57% of the households in Japan [28]. The Yomiuri Shimbun website has 138.84 million page views per month, accessed with computers, tablets, and mobile phones by a wide range in demography, including the ages of college students [35]. Furthermore, articles are also read on portal websites, such as Yahoo! Japan News (https://news.yahoo.co.jp/ (accessed on 3 February 2021)).

We counted the number of articles including the term "SDGs" over a period of time and also according to article type. We also explored in what context SDGs were used. We used K-H Coder 2 (https://khcoder.net/en/ (accessed on 3 February 2021)) to count the terms. The results were translated into English and visualized using a word cloud (https://www.wordclouds.com (accessed on 3 February 2021)). Furthermore, we read all of the articles and coded them into 17 SDGs to explore the frequency of each goal mentioned. To maintain the reliability of the coding, we coded them and corrected them after through multiple verifications.

### 2.2. Questionnaire Surveys

We conducted a two-round questionnaire (see Supplementary Information for the survey forms). Similar to the Delphi method, i.e., an iterative process of data collection using a questionnaire [36], the second-round survey was designed based on the first-round survey results for the same participants. The participants were presented with the first-round survey results when responding to the second-round survey. We chose this approach because the additional information can significantly change the answers respondents, especially given the unprecedented circumstances of COVID-19; they may not be sure how they should behave while the opinions of other participants may be informative. Prior to the surveys, we conducted pretests to check for errors and verify the accuracy and consistency of the questions with college students who did not participate in the main surveys, modifying the surveys accordingly. The first and second rounds were conducted online from June 4 to June 10 and July 30 to August 5 in 2020, respectively. The data were analyzed using STATA (IC16.1 https://www.stata.com (accessed on 3 February 2021)).

### 2.2.1. Participants

The participants were 421 college students registered for an introductory course on social survey analysis in the spring semester of 2020 in the College of Policy Science at a university in Japan. All first-year students in the college were registered for the course. While it is mandatory for them to participate in surveys as part of grading, it was their choice whether their answers were included in the analysis. We emphasized that how they answer the questions would not affect their grades. Students who agreed to use their answers for this research signed an informed consent form.

### 2.2.2. First-Round Questionnaire

The first-round questionnaire comprised 19 questions divided into four sections: the attributes of the respondents, recognition of SDGs, behavioral changes caused by recognizing SDGs, and the impact of COVID-19 on their interest and behavior regarding SDGs. The logos were presented in the questionnaire. We included questions to create index variables for trust and a general concern about sustainability (Table 1). Among the various factors influencing the approval of nudges (e.g., gender, age, political attitudes, types of nudge, and its resonance with the interests and values of individuals), Sunstein et al. [9] recently confirmed the correlation between trust in public institutions and the approval of nudges. As SDGs can be promoted not only by public institutions, but also by other institutions, we included trust in municipalities, the government, and the United Nations, as well as private companies and non-governmental organizations (NPOs/NGOs). We used a 7-point Likert scale from 1 = "do not trust at all" to 7 = "completely trust." Variables for concern about sustainability were based on the variables developed by Grunert et al. [37]. As their variables were focused on the food industry, we modified them and created



thirteen more general and suitable variables for SDGs. Following Grunert et al. [37], we used a 7-point Likert scale from 1 = "only slightly concerned" to 7 = "extremely concerned." The scale reliability of the indices was checked using Cronbach's alpha [38].

**Table 1.** Items used to construct an index variable for concerns about sustainability.

1. Child labor
2. Logging of the rainforest
3. Starvation and malnutrition around the world
4. Pesticides in food production
5. Poor treatment of animals in food production
6. Environmental destruction due to economic activities
7. Food wastage
8. Overuse of natural resources
9. Poor working conditions and low wages in economic activities
10. Single-use or unrecyclable packaging and containers
11. Amount of packaging used
12. Carbon dioxide emissions associated with economic activities
13. Energy consumption associated with economic activities

The recognition of SDGs comprised two questions: whether they had heard of SDGs or seen an SDG logo. We used a 4-point Likert scale from 1 = "very often" to 4 = "not at all."

Behavioral changes due to the recognition of SDGs is key to assessing whether SDGs function as a nudge. We asked the respondents if they had changed their behavior because of this recognition. The respondents who changed their behavior were asked to elaborate on what changes they had made in an open-ended question. We categorized them into seventeen SDGs and counted the frequencies. We also asked the respondents without changes to their behaviors why SDGs did not function as a nudge for them. As the survey was conducted after the COVID-19 outbreak, we emphasized that these questions were in reference to the period before COVID-19.

To measure the impact of COVID-19 on their interest and behavior regarding SDGs, we asked them how the outbreak changed their interest and behavior regarding SDGs on a 5-point scale. To explore the reasons behind their choice, we used an open-ended question, which we coded and cross-checked.

### 2.2.3. Second-Round Questionnaire

The second-round questionnaire was designed after reflecting on the results of the first-round questionnaire in relation to COVID-19. The second-round was conducted approximately two months after the first-round. The questionnaire asked the respondents' intention to change their behavior in response to other students' views on SDGs and COVID-19. The COVID-19 pandemic has yielded changes to young people's behavior, impacting SDGs both positively and negatively. For example, increases in people using disposable masks and throwing them away has led to littering in rivers and oceans, but there has also been decreased carbon emissions and reduced air pollution [39,40]. We expected that the respondents lacked information regarding COVID-19, such that they were unsure of how to behave regarding SDGs in this type of crisis. The respondents' perceptions may change with new information, including peer and group information [41]. Following the Delphi method [36], in the second-round questionnaire, we provided the respondents with the behavioral changes in their peers and their opinions on SDGs and COVID-19, which were reported in the first-round questionnaire. We shared summarized information and open-ended answers in verbatim text.

## 3. Results and Discussion

### *3.1. Permeation of SDGs into Japanese Society before and during COVID-19 (RQ.1 and 3)*

From the Yomiuri database service, we retrieved 734 news articles including the term "SDGs" from January 2015 through June 2020. We did not have access to the main text of 23 out of the 734 articles because they were not available in electronic form. The occurrence of 734 articles is significant because there were only 25 articles that included the term "MDGS," i.e., Millennium Development Goals, in the same database up to June 2020; this demonstrated the prevalence of the term SDGs. Figure 1 shows that few articles included the term "SDGs" at the beginning of the period (2015); however, this increased from around the middle of 2017 until February 2020. The number of articles began to increase because the Japanese Minister of Foreign Affairs participated in a ministerial conference on SDGs held between July 17 and 19, 2017, at the UN. The COVID-19 outbreak, which the media began to more intensely cover at the beginning of February 2020, appeared to lower the public's interest in SDGs. However, instead, there were articles that mentioned both COVID-19 and SDGs. For example, some articles reported on the cancellation or postponing of SDG-related events (e.g., articles on February 22, March 13, and May 21, 2020). Moreover, an article on June 2, 2020 reported that, as part of SDG-related activities, a museum had prepared SDG badges and posters to show appreciation to healthcare workers fighting COVID-19. This indicates that the COVID-19 outbreak led to a decrease and a change in media citations of SDGs, as well as our focus on SDGs.

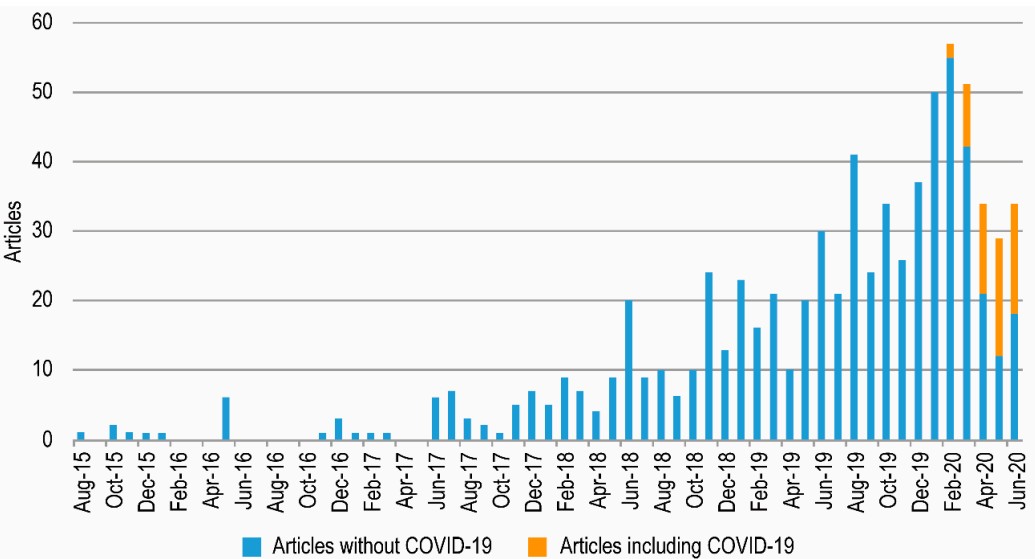

**Figure 1.** The number of articles including the term "SDGs" (Sustainable Development Goals) from August 2015 to June 2020. Articles including both SDGs and COVID-19 are in orange.

Table 2 lists how the articles are spread across sections or categories; all categories except for the sports section. One of the interesting findings is that there was a significant increase in the number of articles in the "regional" section. There were events offering people the opportunity to learn about SDGs (e.g., 13 May 2019, and 20 November 2019). Municipalities adopted SDGs as part of their policies and some established a section for the promotion of SDGs in their office (e.g., 29 May 2019 and 6 August 2019). The Japanese government promoted SDGs by selecting "SDG future cities" [42].

**Table 2.** The number of articles by newspaper section or category.

| Year | Term * | Sport | TV Program/ Entertainment | Front | International | Regional | Politics |
|---|---|---|---|---|---|---|---|
| 2015 | 2nd | | | | 1 | | |
| 2016 | 1st | | | 1 | | 1 | |
| 2016 | 2nd | | | | | | |
| 2017 | 1st | | | | | 4 | 2 |
| 2017 | 2nd | | | | 1 | 9 | 3 |
| 2018 | 1st | | | | | 37 | 1 |
| 2018 | 2nd | 2 | | 4 | 1 | 47 | |
| 2019 | 1st | | 1 | 2 | | 80 | 1 |
| 2019 | 2nd | | 5 | 3 | 3 | 132 | 2 |
| 2020 | 1st | | 3 | 6 | 7 | 154 | 1 |
| Total by section | | 2 | 9 | 16 | 13 | 464 | 10 |

| Year | Term | Culture/Households/ Environment/ Education | Society | Economy | General | Commentary/ Feature | Total by Year |
|---|---|---|---|---|---|---|---|
| 2015 | 2nd | | | | 3 | 1 | 5 |
| 2016 | 1st | | | | 4 | 1 | 7 |
| 2016 | 2nd | | | | 1 | 3 | 4 |
| 2017 | 1st | | | | 3 | | 9 |
| 2017 | 2nd | 4 | | | 7 | 1 | 25 |
| 2018 | 1st | 2 | 2 | 5 | 4 | 3 | 54 |
| 2018 | 2nd | 1 | 6 | 3 | 2 | 6 | 72 |
| 2019 | 1st | 13 | 5 | 8 | 3 | 7 | 120 |
| 2019 | 2nd | 10 | 4 | 5 | 8 | 11 | 183 |
| 2020 | 1st | 33 | 16 | 18 | 2 | 15 | 255 |
| Total by section | | 63 | 33 | 39 | 37 | 48 | 734 |

* 1st half: January to June; 2nd half: July to December.

Figure 2 shows the number of articles by SDGs. The goals were unevenly mentioned. A total of 33.2% of the articles (236 out of 711 valid articles) did not mention any goal, indicating that the articles assumed that readers were aware of SDGs or provided them with a brief description of SDGs without mentioning specific goals. The goals were mentioned 1243 times in total. There were two primary pathways that the goals were mentioned: either listed as an example to explain SDGs or addressed as a focused topic in an article. "Goal 1: No Poverty" (174 articles) was often mentioned as an example of an SDG (e.g., August 2017). This may be because it is the first goal mentioned and is easy for readers to understand. "GOAL 13: Climate Action" (132 articles) was already a well-known serious issue. "GOAL 11: Sustainable Cities and Communities" (128 articles) was often cited as the Japanese government promoted "SDGs future cities" [42]. "GOAL 12: Responsible Consumption and Production" (108 articles) is a familiar topic for readers; for example, it includes recycling. "GOAL 14: Life Below Water" (104 articles) and "GOAL 15: Life on Land" (94 articles) often jointly appeared as environmental protection. The uneven frequencies of the goals mentioned appeared to reflect which topics are familiar or those that have a focus in Japanese society.

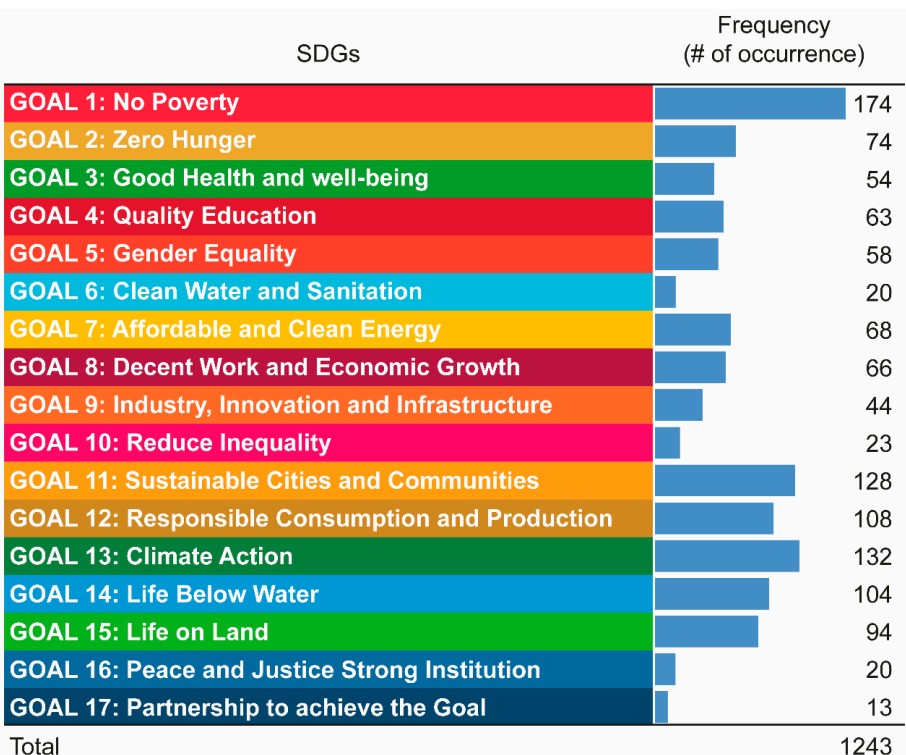

| SDGs | Frequency (# of occurrence) |
|---|---|
| GOAL 1: No Poverty | 174 |
| GOAL 2: Zero Hunger | 74 |
| GOAL 3: Good Health and well-being | 54 |
| GOAL 4: Quality Education | 63 |
| GOAL 5: Gender Equality | 58 |
| GOAL 6: Clean Water and Sanitation | 20 |
| GOAL 7: Affordable and Clean Energy | 68 |
| GOAL 8: Decent Work and Economic Growth | 66 |
| GOAL 9: Industry, Innovation and Infrastructure | 44 |
| GOAL 10: Reduce Inequality | 23 |
| GOAL 11: Sustainable Cities and Communities | 128 |
| GOAL 12: Responsible Consumption and Production | 108 |
| GOAL 13: Climate Action | 132 |
| GOAL 14: Life Below Water | 104 |
| GOAL 15: Life on Land | 94 |
| GOAL 16: Peace and Justice Strong Institution | 20 |
| GOAL 17: Partnership to achieve the Goal | 13 |
| Total | 1243 |

**Figure 2.** The number of articles by SDG. Some articles mentioned multiple goals (n = 711 articles).

Figure 3 is a word cloud where the size of the term is proportional to the frequency of the term appearing in the articles that included the term "SDGs." There are several interesting points worth mentioning. First, "Company" appeared 638 times, indicating the companies involved in SDGs. For example, a bank can simultaneously support business growth and contribute to SDGs. "Region" appeared 615 times, indicating that SDGs permeated the community and regional levels. As previously mentioned, municipalities and prefectural governments have also adopted and promoted SDGs.

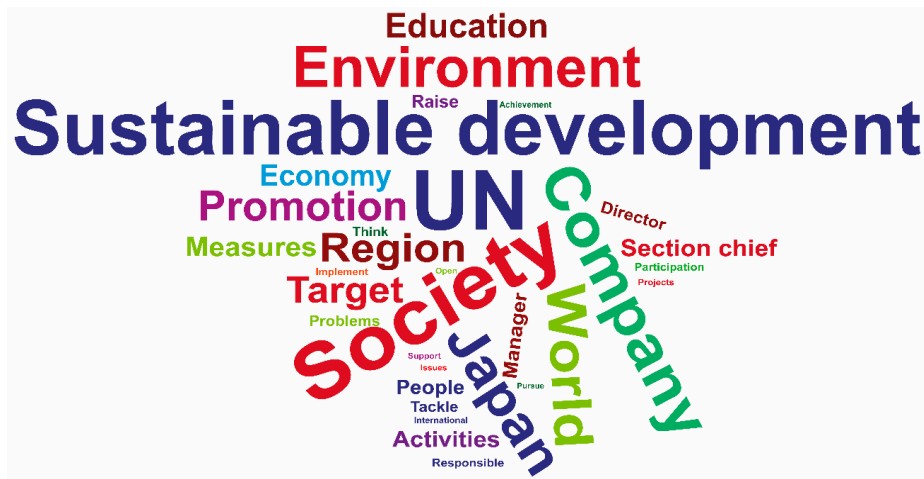

**Figure 3.** Word cloud for the words that appeared with the term "SDGs" in newspaper articles.

### 3.2. Influence of SDGs on Japanese Youth before and during COVID-19 (RQ.2 and 3)

Among the 421 college students registered for an introductory course on social survey analysis, 405 students answered the two rounds and agreed to use their answers for research. Table 3 lists the characteristics of the participants. The gender ratio was similar to that of the college. As the course is for first-year students who were automatically

registered by the office, most respondents were first-year students; therefore, this study was biased toward them. While 48.9% had access to newspaper articles, the rest did not, indicating a limitation for our analysis of the spread of SDGs via the newspaper articles mentioned in the previous section. As expected, younger generations often read newspaper articles on smartphones (45.9%), not on paper (28.7%).

**Table 3.** Characteristics of the survey participants.

|  |  | % |
|---|---|---|
| Gender |  |  |
|  | Male | 61.6 |
|  | Female | 38.4 |
| Year |  |  |
|  | 1st | 90.4 |
|  | 2nd | 7.7 |
|  | 3rd | 0.5 |
|  | 4th and above | 1.5 |
| Frequency of reading newspaper articles |  |  |
|  | Every day | 7.4 |
|  | Almost every day | 8.9 |
|  | Sometimes | 32.6 |
|  | Barely | 28.6 |
|  | Not at all | 22.5 |
| Media to read newspaper articles (multiple answer) |  |  |
|  | Radio | 0.8 |
|  | Newspaper | 28.7 |
|  | Magazine | 0.3 |
|  | PC | 17.5 |
|  | Tablet | 6.2 |
|  | Mobile/Smartphone | 45.9 |
|  | Others | 0.6 |

### 3.2.1. Exposure to SDGs and Resulting Behavioral Changes before COVID-19

The respondents were well exposed to the SDGs. As listed in Table 4, more than 60% of the respondents had been exposed to SDGs for at least some period (67% had heard of SDGs and 64% had seen the logo). This exposure is greater than that of a survey targeted at Japanese citizens aged from 20 to 60, among which 39% of the respondents (n = 519) had heard of SDGs [35]. The results of international surveys on SDGs have revealed that between 28 and 45% of people have heard of SDGs and, consistent with our survey, youth tend to have a higher level of SDG awareness than the average [43].

**Table 4.** Exposure to SDG term and logos.

|  | I Have Heard of SDGs. | | I Have Seen the SDG Logos. | |
|---|---|---|---|---|
|  | **Freq.** | **Percent** | **Freq.** | **Percent** |
| 1. Very often | 117 | 29% | 131 | 32% |
| 2. Sometimes | 153 | 38% | 130 | 32% |
| 3. Rarely | 43 | 11% | 73 | 18% |
| 4. Not at all | 92 | 23% | 71 | 18% |
| Total | 405 | 100% | 405 | 100% |

As the experiences of hearing the term and seeing its logos are highly correlated (0.7357, $p < 0.000$), we generated a composite variable labeled "exposure to SDGs" by taking their mean value to explore the factors affecting the respondents' decisions to change their behaviors.

Table 5 lists the behavioral changes due to exposure to SDGs. We asked the respondents, "When you learned about SDGs, did you change your behavior in any way?" Thus,

this was a self-reported behavioral change. Out of the 279 respondents who had heard of SDGs and/or seen the SDGs logos, 25.4% had changed their behaviors.

**Table 5.** Behavioral changes due to exposure to SDGs.

|  | Freq. | Percent |
|---|---|---|
| 1. Made behavioral changes | 71 | 25.4% |
| 2. Made no changes | 208 | 74.6% |
| Total | 279 | 100% |

Therefore, the question is whether 25.4% is relatively large and meaningful; in other words, can SDGs be qualified as a nudge in terms of their influence to steer people's behavior toward a sustainable society while still allowing them to make their own choices. There have been international surveys on the approval/disapproval of nudges as a policy tool, reporting that approval ranges from 14% to 90% [14,17]. Based on this, 25.4% is in the lower range. However, this requires careful interpretation. First, whether people approve a nudge is one thing; whether the nudge is effective is another. Second, as approval is not the same as people's actual behavioral change, but only their willingness to accept the use of nudges, the corresponding behavioral changes may be lower. For example, the link between pro-environmental attitudes and behavior is not always clear [44]. Willingness to pay for environmental changes, as measured by the contingent valuation method, has been reported to be higher than the actual payment [45–47].

Using an open-ended question, we asked the respondents what changes they had made and categorized them into seventeen SDGs. The results show that 47% of the behavioral changes related to Goal 12: Responsible Consumption and Production. This is reasonable because Goal 12 is relatively easy for respondents who are college-aged students. Figure 4 shows the behavioral changes that the respondents made after their exposure to SDGs. For example, one answered "When I go to a convenience store or a supermarket, I try to carry my own shopping bag so that I avoid getting a plastic shopping bag as much as possible." Overall, there does not appear to be a clear correlation between the frequency of SDGs mentioned in the news articles and the respondents' behavioral changes, as shown in Figure 4. As a direct source of information for their behavioral changes, some mentioned news articles, such that there may be some correlation. Therefore, while the content analysis of news articles is still valid for revealing how SDGs have permeated Japanese society, the weak association is the limitation in this study to explain how the media has changed their behaviors. Future studies must critically analyze broader information sources, including tv programs, social network services (SNS), and education on sustainable development (ESD) [48]. Our survey revealed that ESD is quite influential as some of the respondents mentioned that learning about SDGs in class was the direct source of information that motivated them to change their behaviors. As shown in Figure 4, what people can do is relatively limited (e.g., younger generations in this case). Those providing ESD should place more significance on the goals that are important in their context.

We asked the respondents who did not make any changes (74.6% as lists in Table 5) for their reasons, which are listed in Table 6. Thirty percent said that the information conveyed by SDGs was not convincing enough to change their behavior. Twenty-eight percent answered that SDGs are not simple enough to understand.

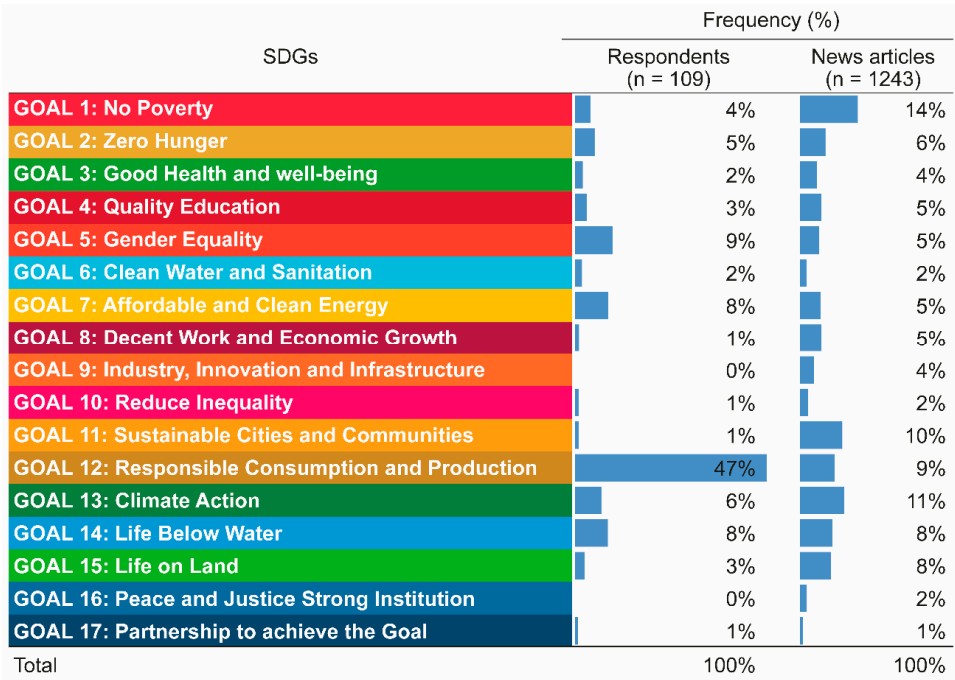

**Figure 4.** The respondents' behavioral changes due to their exposure to SDGs in comparison with the frequency of SDGs mentioned in the news articles. n is the number of occurrences in the questionnaire and news articles.

**Table 6.** Reasons for not making any behavioral changes (multiple answers).

| Reason | Freq. | % |
|---|---|---|
| 1. Even after I heard of SDGs, I was not convinced to change my behavior. | 82 | 30% |
| 2. I did not really understand SDGs. | 75 | 28% |
| 3. I did not really think making the changes was a good idea. | 4 | 1% |
| 4. Even if I made changes, I would not gain anything from it (money or benefits). | 18 | 7% |
| 5. It was too hard to make changes (in terms of time and/or money). | 29 | 11% |
| 6. Other | 61 | 23% |
| Total | 269 | 100% |

Table 7 lists the logit model used to explore the predictors of behavioral change. The dependent variable was 1 if a respondent made changes and 0 if not; the respondents who answered that they did not know about SDGs (Table 6) were excluded to focus on the impact that SDGs have on behaviors. The results show that females and seniors (2nd year and above) are more likely to change their behaviors. In the pro-environmental behavior literature, while some studies support gender differences and ages [41,47], this is not always the case [49]. Furthermore, many current studies on pro-sustainability behaviors involve more than pro-environmental behaviors. They are sometimes used interchangeably, but not identically, as the former embraces more than the environment. Although their focus was on attitudes toward public support for SDGs, not behavioral changes, Guan et al. [50] found that male, younger cohorts, and better educated respondents were more likely to support SDGs. Our findings were not consistent with their results in that our results show that female students are more likely to support SDGs. Owing to the negligible literature, further studies explaining these predictors are needed. The frequency of newspaper reading is not statistically significant. Concerns over sustainability and exposure to SDGs are statistically significant. The latter is consistent with Guan et al. [50], who found that knowledge on SDGs is associated with support for SDGs. Trust in organizations was not statistically significant, which was not consistent with international surveys [14,17]. Other than the possibility that trust has nothing to do with behavioral changes, this may be due to

contextual differences. The current study is about Japan, which was not included in those surveys and also includes organizations other than the government that promote SDGs.

**Table 7.** Logit model to explain the predictors that influenced respondents' behavioral changes due to exposure to SDGs.

| Variable | Description | Coefficient | Std. Err. | *t*-Stat | *p*-Value |
|---|---|---|---|---|---|
| Gender | 1: Male, 2: Female | 0.951 | 0.319 | 2.98 | *** |
| Year | 1: 1st year, . . . , 5: 5th or above year | 0.589 | 0.274 | 2.15 | ** |
| Newspaper read | 1: Everyday, . . . , 5: Not at all | 0.073 | 0.140 | 0.52 | |
| Sustainability | Index variable, 1: Least concerned, . . . , 7: Most concerned | 0.042 | 0.013 | 3.17 | *** |
| Trust | Index variable, 1: No trust, . . . , 7: Perfect trust | 0.029 | 0.034 | 0.84 | |
| Exposure to SDGs | Index variable, 1: very often, . . . , 4: Not at all | −1.287 | 0.136 | −4.72 | *** |
| Constant | | −4.796 | 1.522 | −3.15 | *** |
| *N* | | 278 | | | |
| Log-likelihood | | −127.121 | | | |
| Pseudo R$^2$ | | 0.195 | | | |

\* $p < 0.10$; ** $p < 0.05$; *** $p < 0.01$.

### 3.2.2. Impact of COVID-19 on Interest, Behavior, and Behavioral Intention

Given the unprecedented challenges posed by COVID-19, understanding how COVID-19 impacts people's interest, behavior, and behavioral intention is critical. To be more specific, when people are facing COVID-19, do SDGs still function as a nudge?

Table 8 lists how the respondents' interests, behaviors, and behavioral intentions changed after the COVID-19 outbreak. The survey of behavioral intention in the second round is discussed later.

**Table 8.** Impact of COVID-19 on respondents' interests, behaviors, and behavioral intentions for respondents who had changed and did not change their behavior before COVID-19.

| | | Interest * | | | Behavior ** | | | Behavioral Intention after Listening to Other Respondents' Opinions *** | | |
|---|---|---|---|---|---|---|---|---|---|---|
| | | Up | Unchanged | Down | Up | Unchanged | Down | Up | Unchanged | Down |
| Behavioral change due to exposure to SDGs | Yes | 33 | 34 | 4 | 17 | 48 | 6 | 66 | 5 | 0 |
| | | 46% | 48% | 6% | 24% | 68% | 8% | 93% | 7% | 0% |
| | No | 58 | 144 | 3 | 13 | 186 | 6 | 170 | 32 | 1 |
| | | 28% | 70% | 1% | 6% | 91% | 3% | 84% | 16% | 0% |
| | Total | 91 | 178 | 7 | 30 | 234 | 12 | 236 | 37 | 1 |
| | | 33% | 64% | 3% | 11% | 85% | 4% | 86% | 14% | 0% |
| Fisher's exact test | | 0.001 | | | <0.001 | | | 0.11 | | |

\* up (increased significantly and increased); down (decreased and decreased significantly); ** up (very proactive and proactive); down (less proactive and almost no actions anymore); *** up (more proactive and proactive); and down (less proactive and much less proactive).

In contrast to concerns over the negative impacts of COVID-19 on SDGs [21] and some reports regarding the negative impacts of COVID-19 on the environment (e.g., discarding masks and gloves into the ocean [40]), the results show that most respondents who were exposed to SDGs maintained or even strengthened their interest (64 and 33%, respectively) and behavior (85 and 11%, respectively). This is an encouraging finding for the United Nations that claims COVID-19 is an unprecedented wake-up call and a real opportunity to make a profound systematic shift to a more sustainable economy [26].

The impacts on interest and behavior were different between respondents who had changed their behavior with exposure to SDGs and others who did not, at statistically significant levels (Fisher's exact tests: $p = 0.001$ and <0.001 for interest and behavior, respectively). In other words, the respondents with whom nudging by SDGs worked before COVID-19 had more of a proactive change than those who were not nudged. This

difference is reasonable because people who had previously engaged in SDGs should have a stronger association with SDGs.

The finding showing a rather positive influence was unexpected, but the reasons revealed the logic behind the finding, as shown in Figure 5a–d. We grouped the answers to the open-ended question to count the frequencies of similar answers.

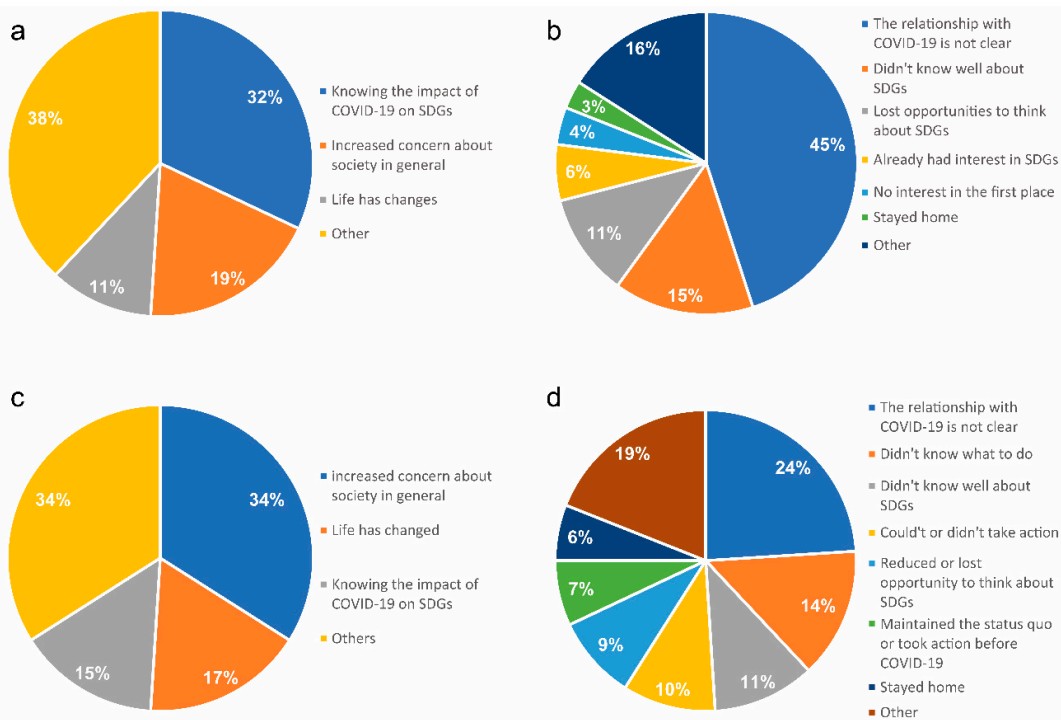

**Figure 5.** (**a**) Reasons that respondents raised their interest in SDGs under COVID-19 (n = 139). (**b**) Reasons for maintaining their interest in SDGs under COVID-19 (n = 205). (**c**) Reasons for adapting their behaviors with respect to SDGs under COVID-19 (n = 35). (**d**) Reasons for maintaining their behaviors with respect to SDGs under COVID-19 (n = 270).

For respondents who strengthened their interest and behavior, COVID-19 was a prime opportunity, as claimed by the United Nations [26]. There were three primary reasons for strengthening their interest and behavior: "Knowing the impact of COVID-19 on SDGs," "Increased concern about society in general," and "Life has changed." One respondent answered, "I saw an article in the newspaper that COVID-19 slowed economic activities for a short period of time, easily resulting in achieving some percentage of the target $CO_2$ reduction and increasing plastic waste." "Increased concern about society in general" includes, for example, "I wanted to do something that would be useful to society." Another respondent said, "I felt that it was necessary to take action on the SDGs so that I can continue my normal life in the future."

Respondents whose interest and behavior did not change appear to comprise two groups: those who lacked the understanding of how to deal with SDGs under COVID-19 (e.g., "The relationship with COVID-19 is not clear," "Didn't know enough about SDGs," and "Didn't know what to do") and the other who maintained their interest and behavior identical to that before COVID-19. People are understandably confused with respect to the right actions when faced with an unprecedented situation. Therefore, we hypothesized that people can learn from each other in such a situation.

Perceptions can change even through informal information from peers [41]. To test this, we shared their peers' reasons for their changes in interest and behavior, collected in the first-round survey. Then, in the second-round survey, we asked the respondents to answer if they would like to change their behavior (i.e., behavioral intention). As listed in Table 8, both groups (i.e., those who changed their behavior due to exposure to SDGs and those who did not) had a strong intention to behave more proactively (93%

and 84%, respectively), and there was no statistically significant difference. Furthermore, respondents answered that their peers' opinions were very helpful (21.9%) or somewhat helpful (55.5%), as listed in Table 9, indicating the importance that the opinions of peers have under such unprecedented circumstances.

**Table 9.** Helpfulness of peers' opinions in determining behavioral intention.

|  | **Freq.** | **%** |
|---|---|---|
| 1. Yes, their opinions were very helpful. | 60 | 21.9% |
| 2. Yes, their opinions were somewhat helpful. | 152 | 55.5% |
| 3. Cannot say either way. | 44 | 16.1% |
| 4. No, their opinions were not very helpful. | 14 | 5.1% |
| 5. No, their opinions were not helpful at all. | 4 | 1.5% |
| Total | 274 | 100% |

Following these results, did SDGs function as a nudge? Figure 6 summarizes our findings. Among the respondents who were exposed to the term SDGs or their logos (68.9%), 25.4% changed their behaviors (i.e., were nudged) toward SDGs. The COVID-19 outbreak impacted some of them positively or negatively. Interestingly, after the COVID-19 outbreak, the number of respondents who changed their behaviors slightly increased from 25.4% (71) to 28.3% (79). However, owing to the limited sample size, this difference may not be conclusive. It is also interesting that some respondents who did not change their behavior before the COVID-19 outbreak changed their behavior after, indicating that COVID-19 acted as a catalyst to make SDGs act as a nudge. Although COVID-19 is an external shock that cannot be implemented by policymakers to promote SDGs, there is an important implication for policymakers. As knowing the importance of caring about society strengthened some respondents' interest and behavior regarding SDGs, measures that educate people on taking care of society in general (in addition to teaching SDGs) can be effective.

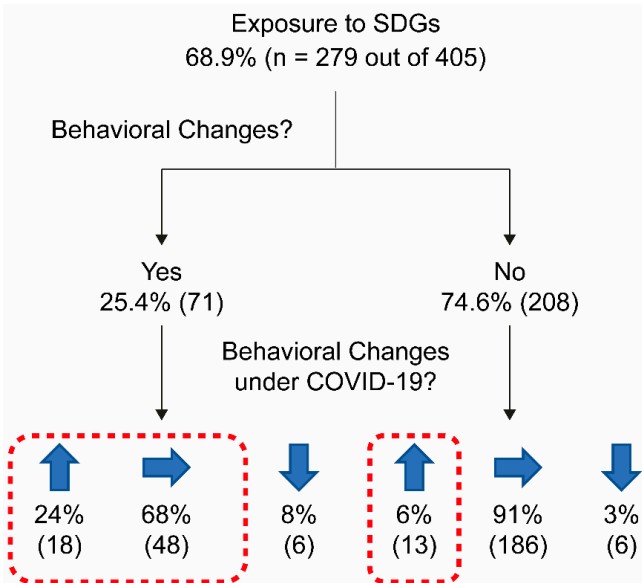

**Figure 6.** Summary diagram for the percentages of respondents nudged by SDGs before and after the COVID-19 outbreak. The large blue arrows (up, right, and down) indicate behavioral changes (very proactive and proactive, no change, less proactive, and almost no actions anymore, respectively). The respondents in the red-dashed circles are speculated to have been nudged.

## 4. Conclusions

At present, SDGs and their logos appear to be ubiquitous in Japanese society. As they share certain characteristics with the concept of nudges, SDGs can qualify as a nudge. For example, SDGs are not legally binding, involve highly complex issues, and are difficult to receive prompt feedback for actions. This is consistent with the type 2 nudge based on the level of intrusion of information campaigns, steering individuals' behavior to achieve a desired collective end [10,17], and promotion by various actors. To our knowledge, this is the first study to investigate whether SDGs can be qualified as a nudge. This study focused on whether SDGs and their logos steer respondents' behavior toward achieving SDGs without forcing them to do so.

The content analysis of newspaper articles revealed that the number of articles including SDGs had increased rapidly up to the COVID-19 outbreak, which tempered the spread of SDGs in the newspaper. Permeation was notable given that while there were 734 articles that included SDGs, there were only 25 articles that included MDGs, a predecessor to SDGs. The topic has spread from the general section to others, such as the regional, culture, household, environment, and education sections. Goals were not equally addressed, but were biased toward Goals 1 (no poverty), 11 (sustainable cities and communities), 12 (responsible consumption and production), 13 (climate action), 14 (life below water), and 15 (life on land). The word cloud (Figure 3) showed that "company" and "region" are often mentioned in these articles. As they appear to have reflected policies and activities implemented in Japan, they can be a proxy for the permeation of SDGs in Japan.

More than half of the respondents, college students, had heard of SDGs or seen the logos, which was higher than surveys in other countries. Although there are no definite criteria for the degree of behavioral changes that can be qualified as a nudge in terms of its efficacy, exposure to SDGs via the terms and logos certainly influenced some respondents' behavior (25.4%). This indicates that information campaigns to increase the recognition of SDGs can be an effective policy measure. Behavioral changes related to Goal 12 (responsible consumption and production) (47%) were prominent. The relationship between what behaviors students adopted and what was mentioned in the newspapers was not clear, indicating a need for further media analysis including other media (e.g., TV programs and SNS). Contrary to the international survey by Sunstein et al. [14], trust in organizations was not a predictor for supporting SDGs as a nudge. As Japan was not included in their survey, it would be interesting to perform research directly comparable with their results. Overall, the COVID-19 outbreak served to raise students' interest and stimulate their behavioral changes. A recent survey in Japan supports this finding; 43% of the respondents (including ages from 18 to 69 years old) answered that the COVID-19 outbreak allowed them to make positive changes regarding their awareness and behavior on environmental issues [51]. Therefore, this can be a turning point for the government to further promote SDGs as asserted by the UN [26]. Our study also revealed that perceptions and behavioral intentions significantly changed in the second-round survey in which peers' opinions were shared. Under unprecedented circumstances, such as the COVID-19 outbreak, sharing peers' opinions can change young people's perceptions and behavior. In addition to ESD, sharing peers' opinions can expedite making the COVID-19 outbreak an opportunity to shift toward a more sustainable society. Hartley et al. [41] assert the importance of informal peer and group influence for changes to public perceptions.

This study has several limitations. First, Japan may be a unique case, as the recognition of SDGs was higher than other countries [43]. An international survey exploring the possibility of SDGs as a nudge has not yet be completed. Similar research in other countries would allow us to elicit further understanding of the uniqueness and similarities of how SDGs have permeated different cultures before and during COVID-19. As Sunstein et al. [14] revealed, there should be cultural differences in how a nudge acts. Second, the survey was conducted with college students in a single college, which is a limitation on the generalization of the findings. Research including a broader sample has yet to be completed. Third, while our content analysis was focused on newspapers, investigating

emerging media, such as SNS, and social media, such as Twitter [52] is as important. While the content analysis of the newspaper articles was meaningful to show how information about SDGs spreads, it may not be sufficient to analyze how the younger generations are influenced by SDGs and their logos, as newspapers were not the primary source of nudging in our study. This indicates that there should be other influential sources worth investigating. Furthermore, investigating multiple media sources together is critical, such as newspapers, magazines, television programs, and emerging media, as well as ESD. Fourth, although respondents indicated good behavioral intentions after their peers' opinions were shared, there may be a gap between intentions and actual behavioral changes [53]. Therefore, studies on behavior using, for example, laboratory and field observations, are needed [54]. Finally, our study did not reveal the cost effectiveness of promoting SDGs (e.g., in terms of the monetary costs or administrative efforts required [10]). While it is important to investigate this, the evaluation of SDGs does not fit the conventional cost-benefit analysis [55]. Evaluating the cost-effectiveness of SDGs as a nudge may require an innovative approach in future research.

**Supplementary Materials:** The following are available online at https://www.mdpi.com/2071-1050/13/4/1672/s1, Questionnaire survey form: Data used to produce the figures and logit model (Table 7).

**Author Contributions:** Conceptualization, T.U. and R.S.; methodology, T.U.; software, T.U. and R.S.; validation, R.S.; data curation, T.U.; writing—original draft preparation, T.U.; writing—review and editing, R.S.; visualization, T.U.; supervision, T.U.; project administration, T.U.; funding acquisition, T.U. All authors have read and agreed to the published version of the manuscript.

**Funding:** This study was funded by The Japan Society for the Promotion of Science (grant number 18H03432).

**Institutional Review Board Statement:** Ethical review and approval were waived for this study due to the guidelines by the university the authors belong to.

**Informed Consent Statement:** Informed consent was obtained from all subjects involved in the study. Written informed consent was obtained from the patient(s) to publish this paper.

**Data Availability Statement:** Data used to produce the figures and logit model (Table 7) are provided as supplementary information.

**Acknowledgments:** We are grateful to the students for their participation in the survey.

**Conflicts of Interest:** The authors declare no conflict of interest.

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
