# Peer review of "Have Sustainable Development Goal Depictions Functioned as a Nudge for the Younger Generation before and during the COVID-19 Outbreak?"

_sustainability, doi:10.3390/su13041672_

Round 1

Reviewer 1 Report

Purpose of study and results are concisely presented, and research questions have been addressed. I believe that the paper provides an original perspective, namely the assessement of SDGs as possible nudging tool toward people before and during the pandemic. Anyway, I believe that, in its current status, the paper has several flaws that limit its possibilities to be considered for publication. Therefore, I provide here a list of comments about the paper.

First of all, a significant drawback of this work, is the almost non-existence of a previous theoretical development of literature on the subject matter. In fact, the usual Section devoted to the literature background analysis is missing. I do not refer to the literature about nudge as a concept within behavioral economics (that is quite satisfying), but to the literature debate about SDGs implementation, trasformative power, disclosure and so on. Describing the concept of SDGs as potential nudging source is not enough to set up some propositions and try to corroborate them with an empirical study. This is the reason why there the Section devoted to the discussion makes only little reference to previous studies since it is not possible to compare its results with the previous theory. Moreover, a recall to a theoretical framework is only reported in the Conclusion Section as a need: "Because of its complexity, it may require a rigorous analytical framework drawn from, for example, mass communication theory". Of course, this appears as an odd articulation of arguments in a scientific research paper and limits the relvance and the potential implications of the results.

Moreover, I found some confusion since the Introdution (page 1) in describing the analysis context where SDGs are sometimes promoted by governments or supranational organizations sometimes by companies (see also page 4 lines 177-179) . Instead, I believe that companies belong to the category of main recipient of SDGs as they have to address SDGs in order to keep their legitimacy on the markets. Citizens in general are the potential beneficiaries of SDGs as well as assume an active role as they adopt sustainability oriented behaviors. In fact, the authors write later in the Introduction that "Therefore, people and organizations can decide how much they contribute toward the goal".

There is also some confusion when the paper talks about the nudging effect toward people (in general) and to youngsters. This is slighty disorienting for the reader that does not understand the focus of the paper and the motivation to it. In fact, how can we generalize youngsters-based results to people in general?

Furthermore, I understood that the paper bases on the assumption that SDGs act as a nudge, and it aims at demonstrating how the exposure to the SDGs dissemination could affect youngsters behaviour. However, this is not perfectly clear as sometimes in the paper authors assert that they aim to demostrate that SDGs act as a nudge (see for example Conclusions line 465).

Therefore, I think that authors should be more precise and clear in describing the context of their analysis, providing definitions and defining the borders of their analysis. 

With regard to the empirical analysis, I think that the choice to analyze newspapers contents (first part of the analysis) is quite counterproductive for the paper as it appears clearly that newspapers are not the most used vehicle for SDGs awareness of the young students (second part of the analysis). Even if this issue is acknowledged as a limitation of the study (page 11 line 334), it makes the paper appear as a twofold research not perfectly merged.

Authors should also provide research and practical implications of the study with reference to impact on academic research and on regulators policies.

Lastly, the fact that the limitation to the generalizability of the results has been considered as a limitation of the study, probably makes the article less interesting for an international audience.

Minor observations

FIGURE 1 is in Japanese

A language proofreading is suggested.

Reviewer 2 Report

In this interesting and well-written manuscript, the authors describe their survey results that examine whether the SDGs as a logo and terminology have functioned as a nudge for undergraduate students in Japan. The authors sought to determine if the nudge was effective and whether the COVID-19 crisis affected the efficacy of the SDGs as a nudge toward actions supporting sustainability. In addition, the study sought an indication of how widespread was the awareness of SDGs among youth.

There were several significant points that need clarification or change before publication. These are the following:

  • Page 4, lines 174-179: Trust in the source is certainly one of the factors that lead to the positive influence of nudges, but it is not the only factor. Other important factors in nudges and narratives are plausibility associated with the content and its resonance with local values and norms. The latter may be especially important in Japan, but also elsewhere. This
  • Page 5 table 1: Please explain why there are 13 categories used to construct the index variable that are not matched or linked with one or more SDGs. It is not clear how these terms lead to indication of recognition of the SDGs as specific goals, rather than in the sense of SDGs as a broad concept.
  • Page 11, lines 317-320: Nudges are not necessarily explicit, so whether someone approves of nudges as a tactic should be distinguished from whether the particular nudge is effective. Efficacy in the sense of whether the nudge promotes going from intention to action is not substantively addressed in this work, as noted in lines 335-336 on page 11.
  • Page 12, lines 363-365: Since this study was limited to a narrow distribution of undergraduate students, it is unclear in what sense the results of this study are inconsistent with Guan et al.
  • Page 16, lines 446-455: The changes in behavior post-COVID outbreak may have more to do with general anxiety about responding to threats to society, rather than specifically to awareness or exposure to the term or logo for SDGs. This is noted in the end of this section, but is then somewhat contradicted by Figure 7, which attributes the dichotomous split to exposure to SDGs. The meaning associated with the large blue arrows (up, down, right) should be indicated in the caption.

A minor grammatical point for clarity is on page 11, lines 342-344: “It could be effective for those performing ESD to put more weight on the goals, which that people can contribute in their context, rather than educating them about all of the goals evenly.” The grammatical change is to make clear that it refers to contributing to those goals that are important in context. (This is an important point about ESD, in my view). 

Reviewer 3 Report

Dear Authors,

The submitted paper " Have sustainable development goal depictions functioned as a nudge before and during the COVID-19 outbreak?" is addressing an important and interesting topic, therefore thank you very much for your work and the contribution. The article analyses how to assess sustainable development goals as a term or logo have functioned as a nudge or a choice architecture before and during the COVID-19 crisis..

The Authors give good justification of research conducted. Literature is relevant. The introduction section of the paper outlines the problem. Section 2 provides a extensive literature review. The subsequent sections present the overview of the research conducted, results and evaluation, followed by a discussion and conclusions. I find the paper interesting and technically sound.

Generally, paper is well structured, important theoretical and  practical aspects of the examined problem are studied and presented in a clear and consistent manner.  Paper is well positioned in journal’s aim and scope. Thus I recommend accept the paper.

I suggest to remove Figure 1 as it presents well known basic ideas about SDG (can be done at proofreading stage)

Reviewer 4 Report

The presented article is based on a current topic and is absolutely appropriate on the part of the magazine.
The literature research is based on current sources and has been carefully edited.
The methodological procedure is clearly presented and the authors outlined an applicable research design. This leads to concrete results, which the authors have discussed.
There is one major error or ambiguity in the article.

Reviewer 5 Report

The title of the abstract raises some doubts because it is obvious that during COVID-19 the goals will be better achieved and that no research is required. In my subjective opinion, I am not convinced by this statement. I am not convinced by the statement: The recent COVID-19 outbreak has prompted us to ask an additional question: how does epidemic sustainability conservation work? People seem to be worried about the vaccine, about the number of people dying, about survival, and not about sustainability, unfortunately. In my opinion, the combination of sustainable development and COVID -19 does not correspond very well. Questions 1, 2 and 3 are insignificant. The group is unrepresentative - only students from one school enrolled in the course were surveyed, the results cannot be generalized to the entire population. The examination contains limitations and disqualifies them. The conclusions are very obvious and generate nothing new. I'm sorry, but I do not recommend publishing the article.

Round 2

Reviewer 1 Report

I believe that the paper has significantly improved, and author/s addressed my comments to previous version. Good luck!